# Dam Safety History and Practice: Is There Room for Improvement?

**Rodrigo Joaquín Contreras** [1,*] and **Ignacio Escuder-Bueno** [2]

1   Department of Hydraulic Engineering and Environment (DIHMA), Universitat Politècnica de València, 46022 Valencia, Spain
2   Research Institute of Water and Environmental Engineering (IIAMA), Universitat Politècnica de València, 46022 Valencia, Spain; iescuder@hma.upv.es
*   Correspondence: rjcontre@upv.edu.es

**Abstract:** Dams and reservoirs have always been of interest to human beings, playing a crucial role given the importance of securing water for sanitary use, irrigation, navigation, flood control and energy generation, among others. The main focus of this article is to perform a historical review of dam safety practices. For this purpose, the historical periods are divided into homogeneous periods in terms of dam safety paradigms and, following the narrative of this evolution, the paper considers the fundamentals of the two most important conceptual frameworks applied nowadays: the standard-based approach and the risk-informed one. As a matter of fact, after more than 90 years of experience in the application of dam safety assessment techniques and more than 50 years of recognising and studying the implications of human activity for the environment, today, the industry may have sufficient information and knowledge to take dam safety practice to another stage, being this the core of the discussion that follows the historical review.

**Keywords:** dam safety governance; historical review; standard based approaches; risk analysis

## 1. Dams' Industry and Safety Governance

As stated in the International Energy Agency's (IEA) World Energy Outlook 2019 report [1], the most frequently cited primary purposes of dam–reservoir systems are irrigation and power generation. For this reason, dams play an essential role in the current context of climate and global change. Irrigation, water supply and flood control functions account for 46.9 percent of all registered dams. Hydropower, meanwhile, remains the world's largest source of renewable electricity generation, accounting for more than 15.7 percent of global generation from any source and 61.8 percent of generation from renewable energy sources. Such percentages demonstrate the importance that the maintenance and revitalisation of dams as strategic structures to address the climate crisis recognised in the international treaty "the Paris Agreement", promoted by the United Nations Conference on Climate Change, COP21 [2], in 2015, and in force since 2016, has and will continue to have. Figure 1 presents the total number of registered dams in the world and their primary purposes. In turn, Table 1 presents the distribution of such dams by region along with their primary purposes.

The relevance of the role of dams in the context of the climate crisis having been highlighted and recognised, it is clear that it is necessary to understand the current situation in the field of dam engineering. In this sense, financial institutions such as the World Bank, supported by information obtained by the international Commission on Large Dams (ICOLD), have focused on the safety of dams and downstream communities in the context of ageing infrastructure, population growth and changing environmental conditions.

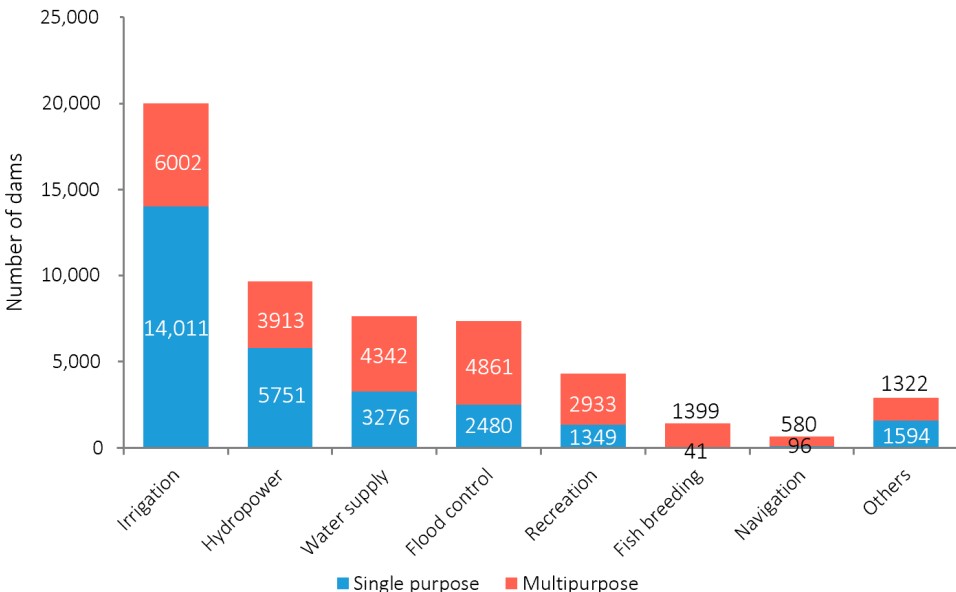

**Figure 1.** Number of registered dams and primary purposes. Source: ICOLD, World Bank Group, 2020 [3].

**Table 1.** Distribution of dams by region and by primary purpose. Source: World Bank Group, 2020 [3].

| Primary Purpose | East Asia and Pacific | Europe and Central Asia | Latin America and the Caribbean | Middle East and North Africa | North America | South Asia | Sub-Saharan Africa | Total |
|---|---|---|---|---|---|---|---|---|
| Irrigation | 7104 | 2192 | 769 | 1032 | 1118 | 4921 | 1133 | 18,269 |
| Hydropower | 1496 | 2447 | 1048 | 51 | 1893 | 170 | 131 | 7236 |
| Water supply | 720 | 1532 | 265 | 138 | 1657 | 65 | 344 | 4721 |
| Flood control | 1023 | 448 | 74 | 135 | 2770 | 4 | 7 | 4461 |
| Others | 50 | 351 | 336 | 119 | 2951 | - | 50 | 3857 |
| No data | 19,198 | 140 | 141 | 32 | 46 | 221 | 196 | 19,974 |
| Total | 29,591 | 7110 | 2633 | 1507 | 10,435 | 5381 | 1861 | 58,518 |

Note: - = not available.

In 2020, the World Bank Group's Laying the Foundations [3] recognised that the world's dams are ageing as the population grows. In particular, a large proportion of the dams registered in ICOLD were built between 1950 and 1989, so that there have been more 19,000 dams in operation in the last 50 years or more (Figure 2). The world's large dams make a significant contribution to the efficient management of water resources, which are unevenly distributed and subject to large seasonal fluctuations. However, with age comes deterioration, so the increasing number and age of large dams, together with changes in downstream demographics and the economic value of the assets, expose a situation that requires increasingly sophisticated management tools and approaches capable of identifying and managing the associated risks.

The World Bank (2020) [3] acknowledges the difficulties of comprehensive safety management due to the lengthy preparations required for dam projects, the changing nature of the international market and the fact that there are few large dam programmes at the national level. The document identifies an upward trend in the construction of new dams over the last 10 years, reflecting a consolidated response across sectors to help address energy, water and food security. However, in many countries, this increase in new dam construction has often moved at a faster pace than the evolution of legal, institutional and regulatory frameworks; as a result, governments may not have sufficient capacity to ensure the management of dam safety.

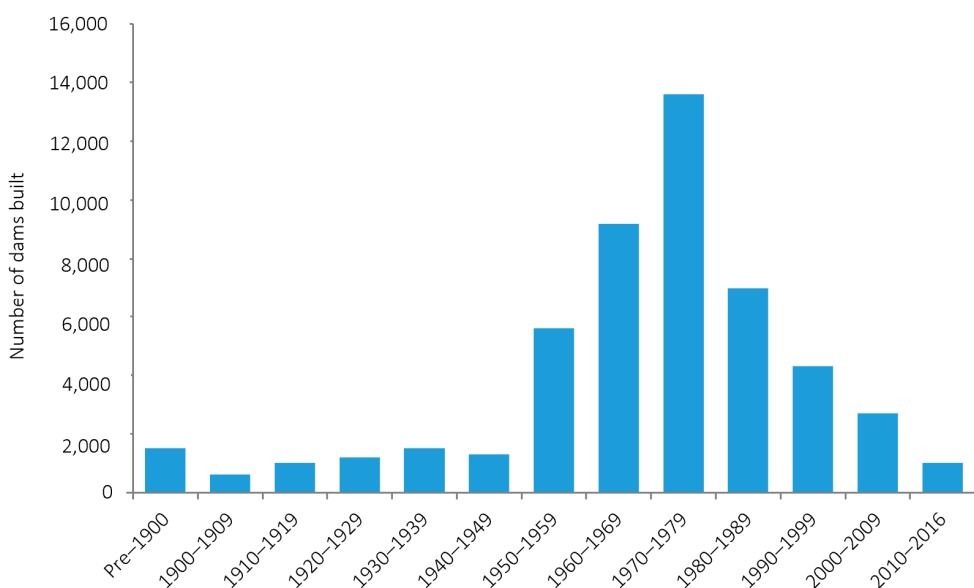

**Figure 2.** Number of dams built and recorded over time. Source: World Bank Group, 2020 [3].

Focusing on the latter topic, the key issue in any dam safety system is to ensure that the institutional capacity is sufficient to meet the expected duty of care. This includes financial resources, human capital and sufficient technical capacity to respond to the challenges of dam management and regulation. To this end, the clear definition of primary responsibilities is a key element of any regulatory framework applied to dam safety governance. Such institutional capacity can be achieved through a framework that includes effective, quality, well-targeted procedures that are conducive to dam safety governance or risk governance. As the World Bank (2020) [3] puts it, well-established governance should provide the necessary funds to sustain the evolution of environmental policies and the underlying understanding of the sector context, including the recognition of changes in hydro-meteorological conditions, downstream population increases and land use changes. These funds are also needed to cope with deterioration due to ageing infrastructure, changes in technical standards and the adoption of improved techniques. The mechanisms for generating such resources are determined by the structure of the owner (public or private), by the type of service provided (hydropower, water supply, irrigation, flood protection, etc.) and by the nature of the monitoring mechanisms (self-regulation or autonomous regulators). This can have a significant impact on the quality of dam safety management and the level of assurance expected.

In this sense, regulatory frameworks for dam safety are commonly financed through a combination of tax-based government revenues and user-generated payments. However, the ability to fully meet expected requirements is still weakened in many parts of the world by imbalances in costs, tariffs, financial demands and limited budgets. According to the survey presented by the World Bank (2020) [3], strategic financial planning, together with tools that facilitate the prioritisation of dam safety measures and resources, can be useful in budget-constrained environments and even more so in the pursuit of the sustainable management of both economic and environmental resources. In this way, the balance between the available financial resources and the efficiency of decision-making processes must be placed in a multi-criteria framework that can match resources with requirements, addressing the wide range of needs.

An additional challenge for dam safety governance arises in the context of the coexistence of different legal and institutional regimes within a transboundary river basin. Such a situation leads to the creation of new sources of risk between different safety standards and obligations. Under these conditions, a minimum level of coordination between the parties is necessary to ensure a minimum level of safety. One way to resolve the differences

is by proposing a single safety governance system that includes measures to improve and facilitate the exchange of information related to operations, to improve the coordination around emergency preparedness and promote internationally recognised principles.

It follows from the above that the desired governance to ensure the safety of dams and downstream communities in the current context of a global climate emergency is one that offers the highest level of assurance. However, this level depends not only on the structural elements and prevailing policy, but also on the capacity to realise the intentions of such governance. From this follows the need for decision-making methodologies and procedures that are able to integrate multi-criteria management strategies, positioning themselves within basin planning and management processes, with the capacity to generate measures that meet agreed environmental commitments at the necessary speed and scale [2,4,5].

## 2. Historical Review of Dam Safety Practices

Throughout the history of dam engineering, it is possible to identify events that have marked the knowledge development and that provide insight into the current state of the art for further diagnosis. Divided into five periods by the authors, the evolution of science and technology applied to dam safety engineering is presented below.

### 2.1. Ancient and Pre-Classic Period, 3000 B.C.–1850

Until the middle of the 19th century, designs were based on empirical knowledge based on the experience acquired in the execution and performance of the works, but fundamentally on that obtained from the failures that occurred. This period begins with the first written references dated around 3000 BC, as is the case of the Jawa Dam on the Rajil River in present-day Jordan. The failures of the Puentes II Dams in 1802 in Spain and Blackbrook I in 1799 and Blackbrook in 1804 in England are documented examples of failures triggered by insufficient technical knowledge.

### 2.2. Classic Period, 1850–1930

The beginning of empirical–theoretical engineering took place with the application of rational mechanics driven by the physical and mathematical advances of the classical period. The studies of elasticity by Hooke, 1678; of frictional materials by Coulomb, 1776; of structural mechanics by Bernoulli, 1782 and Navier, 1839; of flow in continuous media by Darcy, 1856; of thrust in soils by Rankine, 1857; of stress distribution in semi-space by Boussinesq, 1885; of fluid dynamics by Reynolds, 1887; and of the plasticity of soils by Atterberg, 1911, are some of the foundational works. The integration of such experimentally based theories allowed the consolidation of a theoretical framework to understand and respond to the dam failure events that continued to occur. Such is the case of the accidents at the El Habra dam in 1881 and the Bouzey dam in 1895, which highlighted the phenomenon of pore water pressure within the developing theoretical framework. At the same time, the design of the works was also based on the application of the physical–mathematical tools deduced within the new theoretical framework, which was also under continuous revision as required by the scientific method of the time, guided by the philosophy of the mechanistic model.

The incipient application of rational methods and the possibility of predicting the appropriate behaviour of works led to the research and development of the concepts of safety and risk in a quantifiable way. The implementation of the theoretical framework and behavioural records made it possible to determine performance ranges and thus establish design parameters that would become standards. In this way, the safety checks were focused on the design phase and aimed at verifying the application of the theoretical concepts and tools. The documented analysis of the failures of the Gleno Dam in 1923, St Francis Dam in 1928, Granadillar Dam in 1934 and Zerbino Dam in 1935 pointed to the inadequate application of design tools and poor decision-making processes as responsible for the accidents, driving the demand for new knowledge and stricter control procedures to meet the socio-economic demands of the early 1900s.

### 2.3. Modern Period Phase I, 1930–1970

Beginning in the 20th century, the theoretical diagnosis of past failures drove the application of process control through the development of physical models, material design and monitoring. The new knowledge provided by soil mechanics and geotechnical engineering, from the works of Karl Terzaghi published since 1925, the creation of the Delft soil mechanics laboratory in 1934 and the first International Conference on Soil Mechanics and Foundation Engineering ISSMGE in 1936, accompanied those processes of improvement in dam engineering. The application of new design and process control tools allowed the optimisation of structures and the development of new solutions. An iconic case was the design and construction of the Hoover Dam between 1931 and 1936. The analysis and monitoring during construction and in the operation phase made it possible to understand the performance of the structures, focusing on the verification of the design hypotheses and inspection of the general behaviour following the Observational Method proposed by Terzaghi. The methodology combines rational analysis based on theoretical foundations with empirical judgement obtained from practical experience. A. Casagrande presented the method as "Calculated Risk" in the 2nd Terzaghi Lecture in 1964 [6], introducing to the community the concepts of unknown risks and human risks.

The accidents at the Fort Peck Dam in 1938 and Malpasset Dam in 1959 gave rise to new knowledge to be applied to the design and monitoring of the works. As a consequence, the sense of safety was broadened. The application of design standards and their control with instrumentation was accompanied by the creation of the American Society for Testing and Materials (ASTM) in 1902, which formalised the quality controls of materials, and by the advances in Total Quality Management from the proposals of K. Ishikawa from 1949 [7].

The post-war period generated renewed demands for water and energy and brought new technologies in civil construction equipment. Such socio-economic conditions led to a rapid increase in the number of dams built worldwide. Between 1900 and 1949, some 5000 dams were recorded as having been built; however, in the period 1950–1959, more than 5500 new dams were recorded (Figure 2). Dam engineers met this challenge with science-based tools for both the analytical assessment of behavioural deviations and the design of mitigation interventions. The creation of the International Organization for Standardization (ISO) in 1947 and the consolidation of the International Commission on Large Dams (ICOLD), founded in 1928, strengthened the idea of an international community for the collaboration and dissemination of knowledge and standards. In particular, ICOLD also promoted the formation of specialised technical committees for the monitoring and evolution of the working techniques of the different areas of knowledge involved in the conception, design, construction and monitoring of dams.

In this period, dam safety was based on the application of technical–scientific standards for design and the subsequent inspection and verification of design assumptions. This allowed the necessary interventions to be designed in an analytical way. The concept of the standard-based approach (SBA) was born, where the concept of risk is introduced through dam classification schemes to reflect its threatening nature in terms of (a) the severity of the consequences of failure; (b) design loads for unlikely events; and (c) safety coefficients.

The consistency between analytical models and field observations reinforced confidence in the theoretical and technical concepts developed and generated a demand for specific and well-identified knowledge, deepening areas that were already being explored. This expanded knowledge in areas already explored could be realised where the technology was available.

The methods applied to generate new knowledge were associated with the analytical–mechanistic paradigm consolidated from the 17th century onwards. Thus, the areas that were studied in greater depth were materials science, structural analysis and process control methodologies. Areas with a lesser degree of knowledge and development, due to their complex characteristics, were left behind, such as atmospheric sciences, earth sciences and management and administration sciences.

The dissemination of the General Systems Theory proposed by Von Bertalanffy in 1937 [8] provided tools to tackle complex problems, giving rise to the development of new knowledge obtained from systemic–organic paradigms.

The accidents in the Vajont Valley and at Baldwin Hills in 1963, both due to causes hitherto unrelated to the main closure system, demonstrated the need to strengthen dam safety assessments based on the SBA approach, setting the tone for the interests of the time, which were no longer limited to technical issues.

*2.4. Modern Period Phase II, 1970–1995*

The publication of "Lessons from Dam Incidents" in 1974 by ICOLD initiated the formal recording of accidents for dissemination within the community. The accidents of the previous period were joined by those reported at the Lower San Fernando Dam in 1971, Teton in 1976, Kelly Barnes in 1977, Machhu II in 1979 and Tous in 1982. Such events led governments to seek answers in the technical sphere, prompting the development of legislation for the regulation and supervision of operating dams. Reflecting these demands were the actions taken by the British government with the creation of the Health and Safety at Work etc. Act 1974, and the creation of the Health and Safety Executive (HSE) in 1975 and the Reservoirs Act 1975 by the US government with the enactment of the Reclamation Safety of Dams Act of 1978 and the Water Resources Development Act 1986, accompanied by the creation of the Association of State Dam Safety Officials in 1983 and by the newly created Federal Energy Regulatory Commission (FERC) in 1977 and the Federal Emergency Management Agency (FEMA) in 1979. In Spain, the Dirección General de Obras Hidráulicas (Directorate General for Hydraulic Works) in 1982 promoted the modification of the document Instrucción para el Proyecto, Construcción y Explotación de Grandes Presas (Instruction for the Design, Construction and Operation of Large Dams), 1967, directed by the Comisión Permanente de Normas para Grandes Presas (Standing Committee on Standards for Large Dams), a document that was approved in 1996 as Reglamento Técnico sobre Seguridad de Presas y Embalses (Technical Regulations on the Safety of Dams and Reservoirs).

Governments sought to promote a regulatory framework for the tasks carried out in the industry, systematising the necessary controls to comply with the technical standards defined by the SBA approach, supported by the state of the art. The objective of the legislation was aimed at the creation of bodies responsible for the review of dam safety practices, including internal and external reviews, the qualification of responsible personnel, the integration of new technologies, emergency preparedness plans and the review of existing dams. The role of such bodies within the new safety management scheme was to ensure the correct interpretation and application of the standards (SBA) in the design, construction and operation processes, as well as to document and alert on aspects not met by these standards. The figure of the external expert panel emerged in this context as a valuable tool for the management of engineering decisions. The technical result was a reduction in the number of recorded accidents [9].

Technological development and the spread of computer tools from 1980 onwards made it possible to systematise the monitoring task and improve behavioural analyses by combining the response obtained with mathematical models and what was observed in situ. In this period, the importance of monitoring interpretation as the main scientific method applied to the detection of anomalies and investigation of causes became evident. Instrumentation interpretation remains to this day a fundamental source of information on dam safety [9].

The relationship of dams with society was becoming increasingly close, so populations were aware of their benefits and impacts. In addition to the negative effects caused by the flooding of ecosystems, the inundation of areas of cultural interest and the lack of planning for the relocation of populations, there was also the perception of risk. The technical–political decision-making process for the development of projects required the introduction of the socio-environmental variable for the evaluation of the impacts generated. The Report

of the United Nations Conference on the Human Environment, held in Stockholm in 1972 [10], formalised and brought to the global level the socio-environmental problems and difficulties encountered in the planning, construction and operation of dams.

The more than 30 ICOLD Bulletins published between 1970 and 1990, associated with materials technology, design criteria, analysis techniques, monitoring and inspection, safety guidelines and dams and the environment, reflect the interest in consolidating and disseminating the knowledge that would define the technical standards to be followed.

The socio-environmental demands discussed at the Stockholm Conference in 1972 [10], which proposed the application of sustainability policies based on the declaration of principles based on a common global conviction, were joined in 1988 by the concern for climate change with the creation of the Intergovernmental Panel on Climate Change (IPCC), with the support of the World Meteorological Organisation (WMO) and the United Nations Environment Programme (UNEP). Accompanying these processes, the first publications related to risk and dam safety, alternative to the standard-based approach (SBA), although with reduced impacts at the time, came with the proposals of Baecher, Pate and De Neufville in 1980 [11]. These ideas grew out of advances made in reliability and risk management theories initiated in the 1960s for application in modern technologies such as aviation, chemical plants and nuclear power plants. A larger number of publications related to the risk governance paradigm would emerge from the second half of the 1990s onwards.

Thus, the end of the period was characterised by a slowdown in the pace of construction after 30 years of growth (Figure 2), an improvement in the mastery of classical areas of knowledge and the regular application of control processes, all actions that led to a reduction in the number of accidents reported with known causes.

## 2.5. Contemporary Period, 1995–Present

The publication of ICOLD Bulletin 99 in 1995 provided statistical information on the evolution of the most important aspects of dam safety, identifying the phenomena and causes of accidents, which made it possible to direct efforts to address them technically and mitigate them. The main conclusions indicated that the correct application of the standard-based approach (SBA), i.e., the correct application of standard criteria for design, construction and operation, as well as monitoring, inspection and surveillance, led to a reduction in the number of reported accidents and incidents [9], confirming the benefits of the SBA.

The trust scenario achieved allowed the search for new opportunities to improve dam safety management to begin. The question was as follows: how safe should a dam be in the context of the increasing global awareness of socio-environmental risk and climate change impacts? Ageing structures, limited opportunities for new generations of engineers to gain experience in dam design and construction and the demand for transparency and documented safety assessments, together with the trend towards the deregulation and privatisation of dam operations introducing the profit motive, are some of the reasons for the need to complement the traditional engineering approach based on standards (SBA) in dam safety management [12].

The new needs to be covered were beginning to be identified and recorded by the US control agencies created in the 1970s (see Section 2.4) for the regulation and supervision of dam safety. These agencies were responsible for the management of large portfolios of dams, which, built at the beginning of the 20th century, were close to reaching 100 years of age and therefore their so-called design life. The safety reviews conducted by such agencies and the subsequent evaluation of these reviews by ad hoc inter-agency committees and independent review panels showed that, in general, sound practices were used, but they concluded that some management practices for dam safety needed to be improved [13]. These agencies, in their decision-making process for the allocation of technical and/or financial resources, noted the need for tools based on technical information to support and prioritise safety actions for the management of their dams in a context of limited resources.

In this context, Bowles, Anderson and Glover [14] took up the initiative of Baecher, Pate and De Neufville [11] and in 1998 they published a practical methodology for dam safety risk assessment and management in line with the concepts established by the National Research Council of the United States [15] and the Bureau of Reclamation [16]. Larry Von Thun [17] had already outlined a practical methodology in 1996. The application of these techniques was also beginning in Australia and New Zealand with the developments of the University of New South Wales and ANCOLD and accompanied by the publication of the SA/SNZ standards, 1999 [18], in Canada, with the developments of BC Hydro, 1996 [19] and the publication of the CSA standards, 1991, 1993 and 1997 [20–22].

New tools developed based on risk modelling to support decision making were beginning their evolution in the field of dam safety. The standard-based safety approach (SBA) could thus be complemented by the conclusions drawn from risk analysis and assessment (RA) for subsequent risk-informed decision making (RIDM).

The drive to understand the sources, nature and scale of uncertainties associated with safety was promoted by governments such as the United Kingdom, the Netherlands, the United States, Australia/New Zealand and Canada, all interested in risk management in their industries and understanding it as a useful tool to meet the social demand for transparency in technical–political decision-making processes [13]. This trend is expressed in the 2000 report of the World Commission on Dams [23], which called for an in-depth study of this aspect. At the same time, ICOLD asked all its technical committees to identify opportunities to respond to the demand for greater transparency in decision-making processes, considering the risk assessment approach as a possible opportunity. The interest of financing and insurance entities encouraged the creation of working groups to develop methodologies to assess risks based on acceptability criteria and thus allow investment in safety decisions and/or actions.

The guidelines proposed by the Australian National Committee on Large Dams in 2003 [24] for risk assessment were the result of such a context and became the first reference published by an ICOLD member country. Subsequently, the publication of ICOLD Bulletin 130, 2005 [12] announced the benefits and limitations of risk analysis and assessment (RA) for decision making (RIDM), starting the process of the discussion and critical evaluation of the potential and drawbacks of such an approach. Aspects such as transparency in decision making, the evolution towards probabilistic analysis and simulation methods, the extension of reliability analyses, the inclusion of socio-environmental demand factors and climate impacts could be addressed by the methodologies developed under the RA approach and complement the standard-based approach (SBA) in dam safety assessments.

Currently, risk assessment is defined as the process of examining and judging the significance of risk. Therefore, its results are input to the decision-making process but do not determine the final decision.

A risk assessment framework requires the definition of acceptable risk and tolerable risk as a starting point. Thus, following the Health and Safety Executive reference [25], acceptable risk was defined as "the risk that, for the purposes of life or work, all persons who might be affected are prepared to accept assuming that there is no change in the risk control mechanism", and tolerable risk as "the risk that is within a range that society can live with to ensure certain net benefits; this range of risk is not regarded as negligible or as something that can be ignored, but as something that should be kept under review and reduced further however and as long as possible".

In this context, the HSE, 2001 [26], with the intention to change the rather philosophical approach to risk of past decades, proposed the generalised framework for the tolerability of risk (TOR) in 2001, depicted in Figure 3.

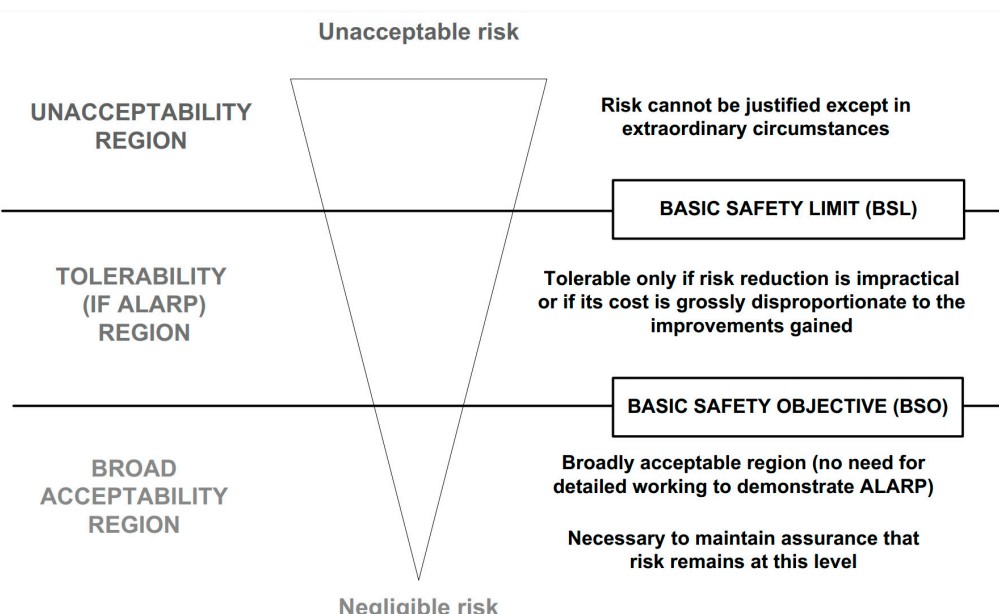

**Figure 3.** HSE framework for Tolerability of Risk (HSE TOR), 2001. Source: ICOLD, Bulletin 154, "Dam Safety Management: Operational phase of the dam life cycle", Paris, 2017.

In order to assess tolerable risk, the developing framework introduced sociological considerations of risk perception [27,28] to explain the division of risk into individual concerns and societal concerns. This approach led to the development of individual and societal risk tolerability criteria, some of which are based on the general principles of equity and efficiency. The work of Morgan and Henrion, 1990 [29] grouped risk assessment criteria into three groups, which the HSE, 2001 [26] refer to as pure criteria and which are based on (1) the principle of equity or individual rights, (2) the principle of efficiency or utility and (3) technology-based. These general principles usually compete with each other and it is necessary to find an appropriate balance between them, resulting in subordinate principles and criteria for the definition of risk tolerability.

It is recognised that the two general principles, equity and efficiency, have broad validity in many societies, spanning a wide range of cultures and political and legal systems. However, as it becomes necessary to move towards subordinate principles, the choices become more complex and are strongly conditioned by social and cultural values and by political and legal systems.

The HSE approach demonstrates that practical approaches to risk assessment often use hybrid tolerability criteria rather than relying on tolerability criteria defined on the basis of a single principle. The aim of the framework is therefore to exploit the advantages of each of the "pure criteria", while avoiding their disadvantages, to resemble the decision process that people use in everyday life and to bring these advantages to dam safety decision making [12]. Thus, the HSE TOR framework uses the equity-based criterion for risks in the unacceptable region and a utility-based criterion for risks in the broad acceptable and tolerable regions.

It is interesting to note that in the definition of tolerable risk, it became common to qualify safety objectives with terms such as "as low as reasonably practicable", known as the ALARP Principle. This principle, understood as the ratio between the cost (time, effort and money) of an additional risk reduction measure and the estimated amount of that risk reduction, led to the introduction of risk reduction indicators.

Among the most widely used risk reduction indicators are the Cost per Statistical Life Saved (CSLS) by Rowe (1977) [30] and the Fatality Prevention Value (VPF) by the HSE.

The case study published by Morales-Torres et. al., 2015 [31] showed that prioritisation sequences based on the results of a given risk model provided adequate and useful information, but they mentioned that other concerns could condition decision-making

processes. The authors showed how the principles of equity and efficiency are captured by each of the risk reduction indicators in the prioritisation process of dam safety investments. Based on these findings, the authors proposed indicators such as the Cost per Statistical Failure Prevented (CSFP), Adjusted Cost per Statistical Failure Prevented (ACSFP) and Equity Weighted Adjusted Cost per Statistical Life Saved (EWACSLS), seeking to balance the principles of equity and efficiency and improve the decision-making process [32].

ICOLD, 2005 [12] acknowledges the effort made but points out that establishing the definitions necessary for the construction of an assessment framework would provide a limited picture given that the full reality includes many other incommensurable and intangible aspects. In this regard, it is noted that the treatment of uncertainties in the estimates made in a risk analysis follows the approach of the HSE, 2001 [26], which recommends the application of the Precautionary Principle. Gullet, 2000 [33] indicates that risk assessment is used to justify the "prevention" of "identifiable threats", while "precaution" aims to avoid uncertain outcomes that may or may not be harmful, although there must be some reason to believe that harm will occur. Thus, the Precautionary Principle justifies acting before knowledge when outcomes are uncertain, i.e., before a perceived threat becomes a known risk.

According to ICOLD, 2005 [12], there is still no common agreement on how the Precautionary Principle should be applied in dealing with uncertainty in dam safety assessments. The work of Morales-Torres et. al, 2019 [34] addressed the effect of unknown uncertainties (epistemic uncertainty) on investment prioritisation based on risk outcomes by proposing coincidence indices. According to the authors, these indices allow the consideration of the desirability of further uncertainty reduction actions as they provide a better understanding of how epistemic uncertainty influences decision making.

It has been identified that it is at the risk assessment stage, and independently of the results of the risk analysis stage (see Section 3), that social, normative, legal, property and other values and value judgements come into play. It is these context-specific factors that control in which region a given risk falls. For this reason, different countries and even different organisations within the same country may classify the same risk differently.

Driven by the work of the HSE, different entities have proposed different guidelines and recommendations for the application of risk assessment and management techniques, such as regulatory bodies [26,35,36], professional bodies [12,37–40] and dam owners [16,41,42], all with the aim of informing risk governance. With the same objectives, countries such as China, India, Brazil, Mexico and Argentina, among others, have initiated the development of their own guidelines and regulations [43–47].

Lessons learned from the accidents at Sayano-Shuashenskaya, 2009; Fundão, 2015; and Feijão, 2019, and the incidents at Oroville, 2017 and Hidroituango, 2018, were related to deficiencies in hydromechanical equipment maintenance, mine waste management standards, design and construction engineering standards and inter-agency safety management. This situation calls for a review of safety management as a whole, and its implementation procedures and working methodologies. It highlights the remaining need to include improvements in addressing human, socio-environmental and climate change impacts.

## 3. SBA and RA Approaches to Support the Decision-Making Process

SBA is based on the hypothesis that physical phenomena can be modelled in prototypes for the study of behaviour. Its foundations emerged from historical developments and precedents, both empirical and theoretical, disseminated, discussed and consolidated by the community over time.

The uncertainties of SBA are associated with the way that its processes are applied, usually implicitly and subjectively. These processes are usually carried out by expert engineers, who apply specific and particular reasonings to each project under study. These reasonings, even when formulated by experts, are not free from the constraints of technical–historical knowledge, programmatic and financial requirements and other, not necessarily technical, boundary conditions.

Another source of uncertainty in the SBA is the way in which its results are presented, associated with global safety margins of binary interpretation. For this reason, the conclusions regarding the safety of the work are strongly dependent on the skills and experience of the technical team involved. Even if the highest technical level is ensured, the conclusions remain binary and are only able to capture the technical aspects in the construction process of the knowledge of the safety of the work.

Although the SBA is conditioned by the uncertainty of the theoretical and empirical model prototypes of behaviour, the variability of the parameters that feed these prototypes, the empirical definition of the safety margins and the binary characteristics of its conclusions, its application made it possible to reduce the number of incidents and accidents recorded by ICOLD in its Bulletin 99, 1995 [9].

Because of the way that it addresses the value of human, social and environmental aspects, the SBA has limitations when used as a decision support methodology to address current demands (Section 1). According to ICOLD Bulletin 61, 1988 [48], dam safety decision making under the SBA approach is based on a design philosophy whose two basic premises are as follows:

- That the structure in conjunction with the foundation and the environment performs its function, without significant deterioration during its design life, under normal operating conditions;
- That the structure does not fail catastrophically in the event of extremely unlikely but possible unfavourable conditions.

The analysis of the SBA approach shows that while it addresses some uncertainties by applying conservative values of the applied loads defined on the basis of the Precautionary Principle and safety factors, other uncertainties are discarded or partially addressed. In particular, it is noted that the Precautionary Principle allows some uncertainties to be addressed under the assumption that any human risk is intolerable, which is tantamount to considering that human life has infinite value. This consideration is crucial in the construction of an effective and efficient single framework for decision making since any decision implies attributing value to life implicitly or explicitly.

The work of Mandeloff, 1988 [49] and Lave et. al., 1990 [50] showed the advantages of making implicit values explicit as a first step towards constructing a single decision-making framework for decisions affecting health and public safety. Dubler, 1996 [51] pointed out that the concept of using thresholds of adverse natural events as a basis for defining public safety criteria is not used in other areas of society. Criteria defined in this way may require greater expenditure on fatality prevention. It was Benson, 1973 [52] who first identified the significant ethical and moral problems of the decision-making framework associated with the SBA approach arising from the imposition of arbitrarily imposed criteria whose implicit message was the virtual non-existence of risk.

Traditional risk management is the result of the application of standards and norms whose requirements depend on the classification of the dam as a function of the potential consequences of its failure or potential hazard. Therefore, the outcome of the safety assessment is always binary: the requirements are either met or not met. Here, the degree of non-compliance is irrelevant as ultimately all deficiencies will have to be corrected in order to reach the standard or meet the norms.

In some cases, these standards do not exist, so the possibility of not knowing about the existence of deficiencies is real. In addition, these deficiencies may not be measurable. Not all aspects of safety can be examined using the observational method and application of standards, as envisaged by the SBA approach.

ICOLD Bulletin 61 [48] of 1988 continues to be a guide for the comprehensive treatment of considerations that should be incorporated into the decision-making process following the SBA approach. However, for the reasons cited above, this approach is limited in its ability to explicitly answer the question of how safe is safe and how it demonstrates the safety achieved. These limitations also make it difficult to assess the impact of introducing

expert opinion into economic cost–benefit analyses and the prioritisation of measures in the decision-making process.

The RA approach is based on the hypothesis that the level of risk of a system can be determined from the representation through prototypes. The study of risk is based on a conceptual framework that proposes a logical cause–effect structure for its estimation and analysis, proposing widespread qualitative and quantitative tools and whose discussion began in 2005 [12] and continues to evolve [53–56].

The RA approach relies heavily on a detailed and exhaustive process of determining the potential failure modes of the infrastructure (PFMA). It is these failure mechanisms that are subsequently modelled following logical cause–effect structures. This approach requires making the technical aspects explicit for categorisation and analysis. In addition to this, through its study of consequences, it has made it possible to include social, environmental and economic aspects that contribute to and condition knowledge about the safety of the dam site.

Thus, the RA approach seeks to provide an analysis scheme capable of capturing new variables and integrating them into a single framework of safety considerations that brings improved information to decision-making processes. The historical review presented in Section 2 shows that risk-based safety management was formulated to complement standard-based safety management.

Ultimately, all safety decisions are risk management decisions where zero risk decisions are not feasible, most of the time not affordable. In this situation, finding a balance between the costs of reducing the risk and the benefits derived from risk reduction is critical.

According to Kaplan, 1997 [57], risk can be understood as the combination of three concepts: what can happen, how likely it is to happen and what are the consequences if it happens. This combination of the probability of events and the associated consequences is the starting point for defining risk [58]. Thus, risk analysis (RA) is presented as a methodology for characterising risk in dams and setting priorities in safety management, as it allows the integration of all existing information on hazards, vulnerability and consequences [59].

In general terms, the RA approach to decision making involves the following four components [12,60].

1.  Risk Analysis: After studying the characteristics of the work and the potential threats, the qualitative phase of the risk analysis begins, in which the potential failure modes are formulated (PFMA) and the conditions and events that must take place for the failure to occur are established. The process is carried out using the brainstorming method guided by a risk analysis specialist. Once the potential failure modes are determined, the probability of failure and the consequences associated with the failure of the dam are estimated semi-quantitatively (SQRA) or quantitatively (QRA). This estimation is done with a risk model that combines the probability of the loads, the probability of dam failure (system response) and the magnitude of the adverse consequences due to dam failure.

2.  Risk Assessment: Risk is estimated as the product of the probability of failure and the associated consequences, a decision is made as to whether the existing risk is tolerable or not (risk categorisation) and recommendations are given for measures to reduce it. The results of the risk analysis are taken and compared with the recommended tolerable risk.

3.  Risk Management: Supported by the risk analysis and risk assessment phases, risk management encompasses activities related to risk-informed decision making to prioritise new studies and risk reduction actions (structural and non-structural) and develop programmes associated with the management of dam portfolios. The risk management process includes the evaluation of environmental, social, cultural, ethical, administrative, political and legal considerations.

4.  Risk Governance: A phase that includes all the actors, policies, roles and procedures related to how risk information is collected, analysed and communicated, and from which management decisions are made. This phase defines how risk assessment

and risk management procedures will be implemented within dam management and regulatory agencies.

As discussed in Section 2, the formulation of acceptability criteria is key in the RA approach. The definition of individual risk acceptability criteria is guided by historical data on the activities being assessed, natural hazards and other sources of risk. The definition of social risk tolerability is more complex, as Ball and Floyd, 1998 [61] point out. Hartford et. al., 2009 [62] show how tolerance criteria have been developed in the past in different countries and in different hazardous industries. In current practice, social risk is characterised by F-N graphs relating cumulative probabilities (F) and numbers of victims (N) in log–log plots, which may vary according to the nature of the application [63–65].

The benefits of risk-informed decision making (RA-RIDM) have been identified and recorded by institutions and authors such as ANCOLD, 2022 [24]; USSD, 2017 [66]; USBR, 2022 [67]; Kumar, Narayan and Reddy, 2018 [68]; and Narayan, Patra and Singh, 2018 [69]. ICOLD, 2005 [12] discusses the RA approach, the current status of its application and the benefits. ICOLD Bulletin 154, 2017 [53] complements this with an emphasis on the operational stage of the dam life cycle.

The RA approach is presented as a theoretical method whose practical/operational application is already being incorporated into the regulation documents of benchmark countries, designed to make the decision-making process efficient and transparent, in order to meet the current demands of society. The application and results of this approach, although still very recent, have already shown that the way to implement it at any scale faces a number of cultural and practical challenges, such as those identified by Escuder and Halpin, 2016 [54], who also bring a number of success cases.

The analysis of the current state of implementation of the RA approach identified the first phases of the approach as the most critical in the whole process [36], where the partial conclusions of the approach depend on (1) the brainstorming method for the determination of the potential failure modes (PFM) and (2) the theoretical–empirical methodologies of mathematical modelling for the estimation of probabilities and consequences.

The brainstorming method [70] is a popular method created to foster group creativity. The characteristics of the method make it difficult to record and quantify the level of simplification or completeness of the model prototype designed for the formulation of the potential failure modes (PFMA), introducing systemic uncertainty to the initial phase of estimating the failure probabilities and consequences. Added to this systemic uncertainty are the uncertainties inherent in the process of estimating the probabilities and consequences for risk calculation. The mechanical modelling of the physical phenomena involved, the mathematical representation and its numerical approximation, the epistemic limitations in the interpretation of the results and the determination of degrees of consequences, among others, are sources that contribute uncertainty to the process.

The recent record of the application of the RA approach demonstrates that the explicit consideration of risk contributes to the establishment of a better rational basis for life safety decision making. However, it has been noted that an effective and efficient decision-making mechanism requires significant effort to be made at the stage of identifying and quantifying all risks present, including physical and human factors. The final report prepared after the Oroville Dam Spillway incident in February 2017 [55] highlighted the vulnerabilities of the RA approach for decision making to the human factor introduced at the individual and inter-institutional levels [56].

## 4. Discussion: Is There Room for Improvement?

It is interesting to note that, in their analyses of the current situation, the statements and needs put forward by ICOLD Bulletin 130, 2005 [12] and Bulletin 154, 2017 [53] belong to the same interpretative dimension from which the SBA and RA approaches have emerged, the classical–mechanistic paradigm. Both approaches conceive of dams at their sites as prototypes with unique characteristics resulting in ad hoc solutions, i.e.,

solutions specifically developed for a particular case and not generalisable or applicable for other purposes.

The classical understanding of dams pertinently leads to the need to always include dam safety engineers in the decision-making process. This classical approach to safety limits the responses that dam engineering can provide to current socio-environmental demands and restricts its ability to develop generic applicable procedures, standards, regulations or control codes.

A methodology for dam safety assessment that acquires systemic and reproducibility characteristics, similar to those required by the scientific method, should provide the necessary process structure to

1. widen the degree of participation of the interested sectors, making the communication processes favoured by a systemic language model more transparent and agile, and
2. arrive at robust convergence results that inform the imbalances of the system in its environment, through models able to reproduce and standardise the experimental phase of analysis, to allow confrontation towards the construction of knowledge.

The authors of this paper propose to explore the study of dam safety from the perspective of General Systems Theory as a possible means of improvement. A dam safety assessment methodology formulated on the basis of systemic interpretation would have the necessary linguistic structure to acquire systemic and reproducibility characteristics. Under these conditions, the methodology would provide alternative and/or complementary responses to current socio-environmental demands.

Exploring the proposed aspects may result in the development of a systemic and reproducible dam safety assessment methodology, in the context of a world that embraces the generation and aggregation of data as has never been possible, as well as the use of artificial intelligence and equivalent tools that will certainly challenge all currently accepted paradigms.

**Author Contributions:** Conceptualisation, I.E.-B. and R.J.C.; methodology, R.J.C.; validation, I.E.-B.; formal analysis, R.J.C. and I.E.-B.; investigation, R.J.C.; resources, R.J.C. and I.E.-B.; data curation, I.E.-B.; writing—original draft preparation, R.J.C.; writing—review and editing, I.E.-B. and R.J.C.; visualisation, R.J.C.; supervision, I.E.-B.; project administration, I.E.-B. and R.J.C. All authors have read and agreed to the published version of the manuscript.

**Funding:** This research received no external funding.

**Conflicts of Interest:** The authors declare no conflict of interest.

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
