# Peer review of "Dam Safety History and Practice: Is There Room for Improvement?"

_infrastructures, doi:10.3390/infrastructures8120171_

Round 1

Reviewer 1 Report

Comments and Suggestions for Authors

This study mainly focused on historical review on the dam safety practices, and sorted out some key issues in this field. Overall, the structure of the article is relatively reasonable and is a valuable research achievement. Specifically, some of the discussions are difficult to understand, and it is recommended that the author refine and enhance the language of the entire text to avoid being broad.

Comments on the Quality of English Language

The language proficiency is commendable, although some discussions are not easy to understand.

Author Response

Rodrigo

Reviewer 2 Report

Comments and Suggestions for Authors

This work proposes an extensive review on the practice history of dam safety in stages, with emphasis on SBA and RS dam safety assessment methods, and the room for improvement is discussed. The logical structure of the paper is clear, the analysis is relatively complete, and it has certain academic value. However, this paper is mostly a general introduction, and lacks critical analysis and comments on the advantages, shortcomings and contributions of previous studies. The conclusions drawn are rather generic and provide little guidance for subsequent dam safety management. Therefore, MAJOR revision has to be done.

Major comments

1. The title is not clear enough, maybe a better title would be "History and Practice of Dam Safety Evaluation".

2. The first section seems to directly quote a large number of contents and conclusions from reference 3, which lack of explanation and cannot draw general conclusions from the original literature.

3. The proportion of dam function in Figure 1 seems to run counter to common sense, as the flood control function should be the primary function of most reservoir dam.

4. The division of the five stages of dam safety practice is inappropriate, and the marking events of each stage (such as dam construction, accident cases, evaluation methods and regulations promulgation, etc.) should be clearly given. At the same time, the phase could be distinguished by flag events rather than integer years. Furthermore, the characteristics of each stage and the promotion of safety management concepts should be clearly explained. The role of contemporary risk assessment and the response to strong human activities and climate and environmental change need to be further discussed.

5. Section 3 introduces SBA and RS methods at great lengths. It is suggested to simplify the general introduction and increase the analysis and comments on the advantages, disadvantages and contributions of each method in the form of charts and graphs. 

6. The discussion in section 4 is vague and general, and there are no good countermeasures were given on how to improve dam safety from the aspects of assessment methods and management.

Minor comments

1. It is suggested to unify the ordinates of FIG. 1 and FIG. 2.

2. The references are not standardized. 

Author Response

Rodrigo

Reviewer 3 Report

Comments and Suggestions for Authors

The paper presents a historical overview of practices in terms of dam safety assessment. This overview focuses on the format of the approaches used, for example: empirical knowledge (<1850), empirical-theoretical engineering (1850-1930), deterministic approach, risk analysis approach (for the last twenty years). The paper also presents the interest of the Standard Based Approach and the Risk Analysis approach in the decision-making process.

A few comments are made below:

With regard to section 2, which presents an historical review of dam safety practices, the authors could mention that a similar observation can be made in civil engineering more generally.

The paper could mention the Eurocodes, which introduce a semi-probabilistic approach. However, large dams do not fall within the scope of the Eurocodes.

For the Modern Period (1970-1995), the paper mentions that the interpretation of monitoring data is the main method for detecting anomalies in the behavior of a dam. Surveillance in the broad sense (visual inspection and interpretation of monitoring measurements) is indeed one of the pillars of dam safety. The paper should specify that the interpretation of monitoring data continues to be of great importance for the safety of dams in the Contemporary Period (1995-present).

L428-430: the authors could add references to countries such as Spain, France, the Netherlands and Canada which are developing approaches to dam risk analysis.

The paper can provide more information on the qualitative and quantitative methods used in dam risk analysis approaches, mentioning, for example, failure mode and effects analysis, preliminary risk analysis, cause/event trees, reliability analysis, expert judgement, .....

The paper should mention some of the limitations associated with the RA-RIDM approach, such as the difficulties in assessing the probability of failure using qualitative approaches (e.g. expert judgement) or quantitative approaches (e.g. reliability analysis), or that there are few real cases integrating a quantitative approach into a risk analysis of a dam.

L566-574: These lines underline the interest of the RA-RIDM, mentioning however that the RA approach is presented as a theoretical method. The paper should mention that several countries are already incorporating the RA-RIDM into their dam safety regulations (this is not just a theoretical approach, but also a practical/operational approach used in real dam cases).

With regard to section 4 (Discussion: is there room for improvement?), the reader expects to see several ways for improvement. However, the proposed perspective presented in this section remains very general and superficial. The paper could give a few specific points on possible areas for improvement or development.

Minor comments:

L25-28: these lines focus solely on hydroelectric dams. The introduction to the article could also mention the case of dams for irrigation. Indeed, irrigation is the most frequent main purpose of large dams in the world, and plays an essential role in the context of climate and global change.

L140: “pore water pressure” seems more appropriate than “interstitial pressure”.

L159 (and several times in the paper): the term "monitoring" seems more appropriate than "auscultation".

L226: at Work etc???

L230-234: add the English translation in brackets.

Author Response

Rodrigo

Reviewer 4 Report

Comments and Suggestions for Authors

This is a robust review of the literature concerning strategies, protocols, and methods deployed for the measurement and assessment of risks, and mechanics that inform decision-makers of when and if a form of intervention is needed. The article is well-structured, and well-written (although there is always room for improvement in English). The historical approach and divisions of time into periods of certain safety mechanisms, informed by the development and evolution of soil and water engineering sciences are interesting. The work reads well, and I have no hesitation in recommending it for publication following minor corrections. Please find attached an annotated copy of the paper that marks some of the editorial improvements that can be made to the paper. Overall, even at its present state, the paper can be published and I am sure will attract widespread interest. 

Comments on the Quality of English Language

Please see my comments earlier. 

Author Response

Rodrigo

Reviewer 5 Report

Comments and Suggestions for Authors

p.1, line 16. Change “risk analysis” to “risk informed”

p. 5, line 159. Not sure what is meant by “auscultation”, perhaps a different word would be better. Instrumentation? (also used in line 166, line 248, line 250)

p. 6, lines 227-230. Omits mention of the 1978 Reclamation Safety of Dams Act, which was arguably the most important piece of dam safety legislation to come out of the United States in the 1970s. Also, ASDSO was not “created” by the U.S. Government and is not an organization thereof. Suggest looking at their website to get a better sense of what they represent.

p. 8, line 305. Change “span” to “design” (also in line 466)

p. 8, lines 301 to 311. This entire paragraph is anecdotal. What “US control agencies” are being referred to? Reference [13] is a FEMA “glossary of terms” document from 2004, which seems out of context.

p. 8, lines 312-319. No mention of the Bureau of Reclamation, which was at the forefront of these developments. A practical methodology was outlined by Larry Von Thun in 1996. Suggest referencing the 1997 Public Protection guidelines (your reference [40]) here as well.

·        JL Von Thun (1996). Risk Assessment of Nambe Falls Dam. Uncertainty in the Geologic Environment from Theory to Practice. ASCE Specialty Conference, Madison, WI.

p. 8, lines 324-327. “The drive to understand ... safety was promoted by governments such as ... the United States ... interested in risk management ... and understanding it as a useful tool to meet the social demand for transparency”. It seems like the 1979 Federal Guidelines for Dam Safety (reprinted in 2004) should be cited to back up this claim. There is no other U.S. Government document prior to 2015 that encourages the use of RIDM. In practice, the use of RIDM within the United States circa 2000 was largely limited to the Bureau of Reclamation, which applied the process solely to its own facilities.

·        "Federal Guidelines for Dam Safety," FEMA P-93. Interagency Committee on Dam Safety. U.S. Department of Homeland Security, Federal Emergency Management Agency, Reprinted January 2004.

p. 10, lines 425 to 428. “Following the HSE TOR framework”. Reclamation 1997 is among the documents cited. How could a 1997 document follow the HSE TOR framework if that framework was not developed until 2001? Also, as a side note, Reclamation does not use (and has not used) the tolerable risk concept. This is explained in the following paper:

Galic D. 2023. The Bureau of Reclamation’s new dam safety public protection guidelines: a departure from internationally accepted practice? Proceedings of the 91st Annual ICOLD Meeting, Gothenburg SE.

p. 10, lines 428-430. Some of these references are over 10 years old, calling into question the meaning of the expression “are developing”.

p. 11, line 471. Change “burdens” to “applied loads”

p. 12, lines 507-508, “still under discussion in the engineering community”. The referenced document is from 2005, which calls into question whether its content is still under discussion. The problems faced by the profession in the early days of RIDM are not necessarily the ones we face now.

p. 12, line 515. “safety of the construction site”. Do you mean “dam site”?

p. 13, lines 551-552. Change “ethnic” to “ethical”

p. 13, lines 566-571. The documents cited are somewhat arbitrary. USSD 2017 is not particularly insightful and is in any case focused on the construction setting. ANCOLD 2003 is no longer current and has since been replaced by ANCOLD 2022. There is no reference in this paragraph to any Bureau of Reclamation documents. The authors would benefit from reading the most recent version of the Reclamation Public Protection guidelines, which were released in December 2022 and can be found at:

·  https://www.usbr.gov/damsafety/documents/ReclamationPublicProtectionGuidelines2022.pdf

p. 14, line 599. Change “making-decision” to “decision-making”

p. 14, lines 623-624. “The authors ... propose to explore the study of dam safety from the perspectives of General Systems Theory as a possible way of improvement”. Many organizations, including BC Hydro, Energiforsk, and Vattenfall are already doing this. How is what the authors are proposing different?

General: The paper provides a thoughtful review of current and historical approaches to dam safety, but the entire thing leads up to an idea (we should start using systems theory) that does not necessarily follow from the premise and that is not in itself that ground-breaking. It would be helpful to present a more detailed outline of what such a process might look like and what specific problems faced by today’s practitioners it may solve. In my experience, the systems approach, while intriguing, is better suited for some applications than others. For example, it is useful if your system is complex enough that you cannot tell a priori what the critical failure mechanisms are, but if you are dealing with an essentially passive structure (e.g., an embankment dam for long-term water storage) with known vulnerabilities (e.g., susceptibility to hydraulic fracturing along the foundation contact), then it is really not that useful. There was a workshop held on the systems approach at the 2023 ICOLD conference in Gothenburg, and a review of the workshop materials by the authors could lead to a more informed discussion.

Comments on the Quality of English Language

Minor English edits suggested

Author Response

Rodrigo

Round 2

Reviewer 2 Report

Comments and Suggestions for Authors

It is suggested to unify the quantity unit of y-axis in FIG. 1 and FIG. 2.

Author Response

Thank you for your suggestion. The Y-axis unit in Figures 1 and 2 has been unified.
